# The *Pseudomonas aeruginosa* RpoH (σ^32^) Regulon and Its Role in Essential Cellular Functions, Starvation Survival, and Antibiotic Tolerance

**DOI:** 10.3390/ijms24021513

**Published:** 2023-01-12

**Authors:** Kerry S. Williamson, Mensur Dlakić, Tatsuya Akiyama, Michael J. Franklin

**Affiliations:** 1Department of Microbiology and Cell Biology, Montana State University, Bozeman, MT 59717, USA; 2Center for Biofilm Engineering, Montana State University, Bozeman, MT 59717, USA

**Keywords:** sigma factor, antibiotic resistance, heat shock response, dormancy, molecular chaperones, proteases

## Abstract

The bacterial heat-shock response is regulated by the alternative sigma factor, σ^32^ (RpoH), which responds to misfolded protein stress and directs the RNA polymerase to the promoters for genes required for protein refolding or degradation. In *P. aeruginosa*, RpoH is essential for viability under laboratory growth conditions. Here, we used a transcriptomics approach to identify the genes of the RpoH regulon, including RpoH-regulated genes that are essential for *P. aeruginosa*. We placed the *rpoH* gene under control of the arabinose-inducible P_BAD_ promoter, then deleted the chromosomal *rpoH* allele. This allowed transcriptomic analysis of the RpoH (σ^32^) regulon following a short up-shift in the cellular concentration of RpoH by arabinose addition, in the absence of a sudden change in temperature. The *P. aeruginosa* ∆*rpoH* (P_BAD_-*rpoH*) strain grew in the absence of arabinose, indicating that some *rpoH* expression occurred without arabinose induction. When arabinose was added, the *rpoH* mRNA abundance of *P. aeruginosa* ∆*rpoH* (P_BAD_-*rpoH*) measured by RT-qPCR increased five-fold within 15 min of arabinose addition. Transcriptome results showed that *P. aeruginosa* genes required for protein repair or degradation are induced by increased RpoH levels, and that many genes essential for *P. aeruginosa* growth are induced by RpoH. Other stress response genes induced by RpoH are involved in damaged nucleic acid repair and in amino acid metabolism. Annotation of the hypothetical proteins under RpoH control included proteins that may play a role in antibiotic resistances and in non-ribosomal peptide synthesis. Phenotypic analysis of *P. aeruginosa* ∆*rpoH* (P_BAD_-*rpoH*) showed that it is impaired in its ability to survive during starvation compared to the wild-type strain. *P. aeruginosa* ∆*rpoH* (P_BAD_-*rpoH*) also had increased sensitivity to aminoglycoside antibiotics, but not to other classes of antibiotics, whether cultured planktonically or in biofilms. The enhanced aminoglycoside sensitivity of the mutant strain may be due to indirect effects, such as the build-up of toxic misfolded proteins, or to the direct effect of genes, such as aminoglycoside acetyl transferases, that are regulated by RpoH. Overall, the results demonstrate that RpoH regulates genes that are essential for viability of *P. aeruginosa*, that it protects *P. aeruginosa* from damage from aminoglycoside antibiotics, and that it is required for survival during nutrient-limiting conditions.

## 1. Introduction

Bacteria must respond to environmental stresses, including exposure to toxic chemicals and reactive oxygen species, as well as sudden changes in osmotic pressure, temperature, or pH. For infectious bacteria, stress response includes exposure to the host immune system, which causes damage to DNA and other macromolecules. In bacteria, stress response pathways have evolved that allow repair or degradation and recycling of the damaged macromolecules. One stress response is the heat-shock response [1,2,3,4]. During heat shock or exposure to protein-damaging chemicals, misfolded proteins accumulate in the cells. The heat-shock response results in production of proteases and molecular chaperones that degrade or repair the damaged proteins. Although the heat shock regulons differ among different species of bacteria, in general, heat-shock induces the production of molecular chaperones and proteases for repair or degradation of damaged proteins [1,3,4]. The master regulator of the heat shock response is the alternative sigma factor, σ^32^ (also termed RpoH) [2]. The cellular concentration of RpoH is regulated at the transcriptional, post-transcriptional, and post-translational levels. Upon a sudden increase in temperature (or other stresses that cause protein misfolding), RpoH is released from titration to its molecular chaperones, activating it to bind the RNA polymerase (RNAP) and direct transcription of genes for protein repair or degradation. The RpoH sigma factor directs the RNAP to promoter regions containing the RpoH consensus promoter sequences, which in *E. coli* and other Proteobacteria contains the conserved −35 consensus sequence (TTGAA) and a more degenerate −10 promoter consensus sequence (CnATnT) [5]. The RpoH regulon has been described for several species of bacteria using transcriptomic approaches, including the gammaproteobacteria *Escherichia coli* [6], *Vibrio cholerae* [7], and *Shewanella oneidensis* [8], and the betaproteobacterium *Neisseria gonorrhoeae* [9]. The alphaproteobacterium *Sinorhizobium meliloti* encodes two RpoH paralogs, which have been characterized [10].

*Pseudomonas aeruginosa* is an opportunistic pathogen associated with acute and chronic infections [11,12,13]. One of the hallmarks of chronic *P. aeruginosa* infections is that they occur in biofilms, where the bacterial cells and their extracellular matrices are bound to a surface, such as host tissue or artificial implant devices [14,15]. As a result, much research has been devoted to characterizing the physiological differences of *P. aeruginosa* growing in biofilms compared to cells growing in planktonic culture, with an emphasis on identifying factors that make *P. aeruginosa* biofilms difficult to clear by antibiotics (e.g., [16,17,18]). An important factor for bacterial biofilm growth is that the cells are physiologically heterogeneous [19,20]. Depending on their location within the biofilm matrix, subpopulations of cells may undergo starvation for oxygen or other nutrients. In a previous study, we compared the transcriptomes of *P. aeruginosa* biofilm cells isolated from different biofilm strata [21]. The results indicated that the dormant antibiotic-tolerant cells had very low abundances of most mRNA transcripts. However, transcripts for RpoH and for genes known to be regulated by RpoH were detectable in the slow-growing cells in the interior of the biofilms. The mRNA for *rpoH* was the 45th most abundant transcript in the interior of the biofilms [21]. IbpA is a molecular chaperone that delivers misfolded proteins to refolding or degradasome complexes on the cell pole [22,23] and is regulated by RpoH. The mRNA for *ibpA* was the second most abundant mRNA transcript in the dormant antibiotic-tolerant subpopulation of *P. aeruginosa* biofilms [21]. In another study, we noted an abundance of *rpoH* and *ibpA* mRNA in biofilms cultured in a diabetic mouse model of chronic biofilm infection for thirty days [24]. Together, these results indicated that in biofilms and in chronic infections, bacteria may be subject to protein misfolding stress, possibly due to factors that mimic heat-shock, including cell aging or damage to cellular structures [22]. The RpoH regulon may be required for these cells to maintain viability during dormancy in biofilms or during chronic infections.

Previous reports have described the transcriptomic responses of *P. aeruginosa* during heat shock [25,26]. In addition, there are reports of the *P. aeruginosa* response to the aminoglycoside tobramycin, which involves the AsrA protease, a heat-shock protein [27]. In the present study, we chose to characterize the RpoH regulon independently of heat shock, since heat shock can induce many physiological changes in the bacteria that are independent of the RpoH regulon [6,9,28]. Our attempts to delete *rpoH* from the wild-type strain *P. aeruginosa* PAO1 were unsuccessful, suggesting that RpoH (or genes regulated by RpoH) is essential for survival under normal laboratory conditions. By characterizing the essential gene set of *P. aeruginosa* PAO1 using a Montecarlo simulation of a saturating transposon library, Turner et al. [29] demonstrated that *rpoH* was essential for *P. aeruginosa* growth under the three conditions tested in that study. Analysis of clinical strains by TnSeq also showed that *rpoH* is essential [30]. The *P. aeruginosa* two-allele transposon library contains one *rpoH* mutant strain (labeled *P. aeruginosa rpoH*::Tn5—39115). However, that transposon insertion is in the 5′ end of the *rpoH* gene and likely does not disrupt the *rpoH* gene or RpoH activity, and therefore is not a true *rpoH* mutant strain. Our failure to delete *rpoH* from the *P. aeruginosa* genome, combined with the published studies on the essential gene set of *P. aeruginosa*, indicate that *rpoH* is essential for viability of *P. aeruginosa*. Interestingly, in some bacteria, such as *Shewanella oneidensis* and *Sinorhizobium meliloti*, RpoH (including both paralogs) can be deleted without affecting cell viability [8,10] whereas in *V. cholerae*, RpoH is essential [7].

In order to characterize the RpoH regulon in *P. aeruginosa*, we used an approach similar to the one used to characterize the *V. cholerae* RpoH regulon [7]. We first cloned *rpoH* behind the arabinose inducible P_BAD_ promoter, then deleted the chromosomal copy of *rpoH* while *rpoH* was expressed *in trans*. Using the *P. aeruginosa* ∆*rpoH* strain with *rpoH* under control of the P_BAD_ promoter, we were able to perform transcriptomic analysis of genes induced by a sudden increase (15 min) in the cellular concentration of RpoH. We compared the RpoH regulon described here with published reports of genes induced by heat shock of *P. aeruginosa* [25,26] and found some, but not complete, overlap of gene sets. In constructing the mutant strain, we also noticed that the ∆*rpoH* mutant strain with *rpoH* expressed *in trans* had increased sensitivity to gentamicin. The *V. cholerae* ∆*rpoH* mutant exhibited increased sensitivity to another aminoglycoside, kanamycin [7]. Therefore, we assayed the phenotypic response of the *P. aeruginosa* ∆*rpoH* (P*_BAD_*-*rpoH*) mutant to aminoglycoside antibiotics and other classes of antibiotics. The results indicated that the *P. aeruginosa* ∆*rpoH* (P*_BAD_*-*rpoH*) strain had increased sensitivity to aminoglycoside antibiotics, but not to other classes of antibiotics. Aminoglycosides were also more effective against *P. aeruginosa* ∆*rpoH* (P*_BAD_*-*rpoH*) cultured in biofilms compared to the wild-type strain. Finally, we show that RpoH is required for optimal survival of *P. aeruginosa* during starvation conditions.

## 2. Results

***rpoH* mRNA is induced by heat shock in *P. aeruginosa* PAO1 and by arabinose induction in *P. aeruginosa ∆rpoH* (P_BAD_-*rpoH*).** In previous studies, we showed that *rpoH* and the molecular chaperone-encoding gene *ibpA* are highly expressed in *P. aeruginosa* biofilms cultured in vitro, and in a mouse chronic wound biofilm model of *P. aeruginosa* infection [21,24]. Those results indicated that *P. aeruginosa* may undergo protein-misfolding stress as a result of biofilm or in vivo growth. To further study the response of *P. aeruginosa* to protein misfolding stress, we characterized the global RpoH regulon of *P. aeruginosa* PAO1. Reports from other research groups have described the transcriptome response of *P. aeruginosa* following heat shock [25,26]. Here, we characterized the RpoH regulon independently of heat shock, by inducing a sudden increase in the cellular concentration of RpoH. Our repeated attempts to generate a *rpoH* deletion mutation by allelic exchange were unsuccessful, suggesting that RpoH is essential. One transposon mutant of *rpoH* is available in the *P. aeruginosa* PAO1 transposon library. However, in that strain, the transposon resides in the 5′ end of the gene, and our results, described below, indicate that that transposon insertion does not disrupt RpoH activity. To construct a chromosomal ∆*rpoH* gene deletion in *P. aeruginosa* PAO1, we first constructed a strain where the *rpoH* gene was under control of the arabinose-inducible P_BAD_ promoter, using the low-copy number plasmid vector, pCF430 [31]. With *rpoH* expressed *in trans* (from plasmid pMF477) we were able to delete the chromosomal *rpoH* allele by allelic exchange, generating strain *P. aeruginosa* PAO1 ∆*rpoH* (P_BAD_-*rpoH*). Although *rpoH* is essential for *P. aeruginosa*, *P. aeruginosa* ∆*rpoH* (P_BAD_-*rpoH*) grew without arabinose addition, indicating that there is sufficient leaky expression [32] of *rpoH* from plasmid pMF477 to support growth even when *rpoH* is not induced. By inducing the expression with arabinose, the cells underwent a sudden increase in the cellular RpoH concentration, mimicking the increase in RpoH levels due to heat shock, but without a sudden change in temperature.

To determine the level of *rpoH* mRNA increase upon arabinose addition, cells were cultured to mid-exponential phase (7.75 h) in minimal medium at 37 °C, then a portion were induced by adding 0.1% arabinose to the medium for 0.25 h. At 8 h, samples were removed from the non-induced control cells and arabinose-treated cells for RNA extraction and purification. Reverse transcription quantitative PCR (RT-qPCR) was performed to determine the *rpoH* mRNA abundances in treated and non-treated cells, and to determine the mRNA abundances of two genes regulated by RpoH. The mRNA from the housekeeping gene PA2966/*acpP* was used as a normalization control. We also used RT-qPCR to compare the effects of heat shock from 37 °C to 45 °C on induction of *rpoH* mRNA in the wild-type strain *P. aeruginosa* PAO1. Figure 1 shows the results of 15 min arabinose exposure to *P. aeruginosa* ∆*rpoH* (P_BAD_-*rpoH*) (Figure 1A) and 15 min of heat shock of wild-type *P. aeruginosa* PAO1 (Figure 1B). *P. aeruginosa* ∆*rpoH* (P_BAD_-*rpoH*) had a five-fold increase in *rpoH* mRNA (*p* = 0.0006) following 15 min of arabinose induction (Figure 1A). The *rpoH* transcript abundance increased three-fold (*p* < 0.0001) after 15 min of heat shock in *P. aeruginosa* PAO1 (Figure 1B). To further test induction of *rpoH*, we characterized the mRNA levels of two genes known to be regulated by heat shock and RpoH—PA3126/*ibpA* and PA4386/*groES*. A gene that is not regulated by RpoH—PA4463/*hpf*, was used as a control. The mRNA levels of both *groES* and *ibpA* increased significantly following arabinose induction. The mRNA level of *ibpA* increased 12-fold (*p* < 0.0001) following 15 min of arabinose addition to *P. aeruginosa* ∆*rpoH* (P_BAD_-*rpoH*) and the *groES* mRNA increased 3.8-fold (*p* = 0.0021) (Figure 1A). The *hpf* mRNA control gene showed no significant change in mRNA abundance following 15 min of arabinose exposure (*p* = 0.09). Temperature shift for 15 min also caused an increase in *groES* and *ibpA* mRNA levels in the wild-type strain (Figure 1B). *ibpA* mRNA abundance increased 232-fold (*p* < 0.0001) due to heat shock, and *groES* increased 39-fold (*p* < 0.0001), while the control gene, *hpf*, showed no change in mRNA abundance due to temperature upshift (*p* = 0.73). Although the results show differences in the level of response of these genes to arabinose induction versus heat shock, both conditions resulted in an increase in the mRNA levels of *rpoH* and RpoH-regulated genes, and no differences in a gene not regulated by RpoH, indicating that induction of *P. aeruginosa* ∆*rpoH* (P_BAD_-*rpoH*) with arabinose can be used to identify genes of the RpoH regulon, independently of heat shock.

To characterize the transcriptomic response to a sudden increase in the cellular concentration of RpoH, we performed Affymetrix microarray-based transcriptomics analysis on *P. aeruginosa* ∆*rpoH* (P_BAD_-*rpoH*) where the cells were cultured in biofilm minimal medium (BMM) to mid-exponential phase and then induced with 0.1% arabinose for 15 min. Scatter plots of the transcriptome response to arabinose addition compared to the non-induced controls are shown in Appendix A. Independent biological replicates of *P. aeruginosa* ∆*rpoH* (P_BAD_-*rpoH*) showed highly reproducible gene expression values, with R^2^ = 0.98 (example in Appendix A.) *P. aeruginosa* ∆*rpoH* (P_BAD_-*rpoH*) had 258 genes differentially expressed (at *p* < 0.05 and > twofold change) due to 15 min of arabinose induction of *rpoH*, with 143 genes having increased mRNA levels and 115 genes having reduced mRNA levels (Appendix A). To determine the effect of arabinose on gene expression of *P. aeruginosa* PAO1, we compared the genes differentially regulated by increased *rpoH* expression with our unpublished data on the effect arabinose on *P. aeruginosa*. Of the *rpoH* induced genes, only two genes were moderately induced (less than three-fold) by arabinose alone (*ibpA* and *bfrB*).

As with the RT-qPCR results, the transcriptomics results showed an increase in *rpoH* mRNA levels following 15 min of arabinose induction (16.6-fold, *p* = 0.0014). Analysis by DAVID showed that the functional groups most affected by an increase in RpoH included genes involved in oxidative phosphorylation, stress response, aerobic respiration, and chaperones (Table 1). Functional classes of down-regulated genes include genes regulated by FecR, AMP-binding enzymes, and TonB-dependent receptors. The genes that were down regulated are shown in Appendix A.

**Transcriptome analysis of the *P. aeruginosa* RpoH regulon: regulation of proteases and chaperones.** The transcriptomics results of *P. aeruginosa* ∆*rpoH* (P_BAD_-*rpoH*) showed two- to 16-fold increased mRNA abundance of 29 proteases and chaperones and their accessory proteins, following arabinose exposure (Table 2). Ten of these RpoH-regulated genes are described as essential under at least one growth condition by Turner et al. [29] (shown as “+” in Table 2), who used a transposon library to define the essential gene set of *P. aeruginosa*. The essential membrane-bound protease involved in cell division, PA4751/*ftsH*, known to be induced by RpoH in other organisms, was induced by an increased cellular level of RpoH, both here and by heat shock [25]. The results of RpoH regulation of chaperones and proteases are consistent with the heat-shock responses of other bacteria, where enzymes involved in protein repair and degradation are regulated by RpoH. In addition, seven of the protease and chaperone genes had promoters with consensus sequences similar to RpoH promoters in other bacteria, with a −35 consensus sequence—TTGAA and a −10 consensus sequence—ccATcT. We compared the results from Table 2 with the results of Chan et al. [25] who performed an RNAseq based study of *P. aeruginosa* PAO1 following heat shock from 37 °C to 46 °C. Sixteen of the proteases, chaperones, or accessory proteins shown in Table 2 overlap with the heat-shock response described by Chan et al. (shown in bold in Table 2), with 24 total genes overlapping between the two studies. The overlapping proteases and chaperones between this study and the heat shock study are PA0779/*asrA*, PA1596/*htpG*, PA1802/*clpX*, PA1803/*lon*, PA4542/*clpB*, PA4751/*ftsH*, PA4759/*dapB*, PA4760/*dnaJ*, PA4761/*dnaK*, PA4762/*grpE*, PA4942/*hflK*, PA5053/*hslV*, PA5054/*hslU*, and PA4474, PA4870, and PA4943. STRING network analysis [33] identified a tight cluster of the proteases and chaperones among upregulated genes (cluster #2 in Appendix A). Although highly induced by heat shock in our RT-qPCR experiments, the *ibpA* and *groES* transcripts were not observed to be induced by heat-shock by Chan et al. [25] using RNAseq. Other proteases and chaperones identified here as highly induced following increased *rpoH* expression, but not identified as induced by heat shock, include PA0778/*icp*, PA1805/*ppiD*, PA2830/*htpX*, PA3257/*algO,* PA4472/*pmbA*, and PA4756/*carB* (Table 2).

**RpoH regulation of amino acid and nucleic acid metabolism.** In addition to the chaperones and proteases, several genes, which encode proteins likely involved in protein synthesis or recycling, are also induced by a sudden increase in RpoH levels (Table 3). These include genes involved in proline metabolism (PA0392-PA0394, PA2776). Genes involved in assimilatory nitrate reduction (PA1779-1783), which are a group of proteins that forms a functionally linked cluster by STRING analysis (cluster #6 in Appendix A) are also induced.

From the results here, heat shock likely causes damage to nucleic acids. A sudden increase in RpoH levels caused induction of several *P. aeruginosa* genes involved in nucleic acid metabolism and repair (Table 4). Genes highly induced by increased RpoH levels include PA3272, an ATP-dependent helicase, PA4971, an NTP pyrophosphohydrolase, and its adjacent genes, PA4970 (which appears to have a novel protein fold) and PA4969/*cpdA*, a 3′,5′-cyclic-AMP phosphodiesterase. The PA3011/*topA* topoisomerase and the adjacent nucleotide binding protein PA3010 are also induced by RpoH. *topA* was also observed to be induced by heat shock [25]. *topA* and *cpdA* are classified as essential [29]. The mRNA for the RNA chaperone *hfq* showed a 2.1-fold increased response to induction of RpoH (*p* = 0.0034).

**Transcriptome analysis of the *P. aeruginosa* RpoH regulon: annotated and hypothetical proteins of the RpoH regulon.** Annotated genes regulated by RpoH include the anti-sigma factors PA0763/*mucA* and PA0764/*mucB* for another stress-response sigma factor, σ^e^ (also called AlgT and AlgU) (Table 5). *mucA* and *mucB* were moderately but significantly induced by increased RpoH levels. The PA4749/*glmM* and PA4750/*folP* genes, encoding phosphoglucosamine mutase and dihydropteroate synthase, which are adjacent to and likely on the same operon as the heat-shock protein *ftsH,* were significantly induced by an RpoH increase. Genes of the *ftsH* operon (PA4749-PA4751) are all classified as essential in *P. aeruginosa* [29].

Many genes involved in cytochrome metabolism and respiration are affected by RpoH levels. Cytochrome oxidases and putative cytochrome encoding genes induced by RpoH include PA1318/*cyoB*, PA1319/*cyoC*, PA1555/*ccoP2*, PA1557/*ccoN2*, PA4429 (a cytochrome c1 precursor), PA4430 (cytochrome b), and the hypothetical protein PA2481 (Cytochrome c oxidase, cbb3-type, subunit P). A large operon associated with NADH metabolism, PA2637-PA2649 (*nouA*-*nouN*), is significantly upregulated by an increase in RpoH levels (cluster #4 in Appendix A).

Many genes coding for hypothetical proteins are induced by RpoH. We analyzed publicly available AlphaFold2 [34] models to annotate the function of hypothetical proteins that are induced by a sudden increase in the level of RpoH. Among the highly induced hypothetical proteins are an operon from PA0744-PA0747, which is induced 8- to 14-fold following a 15 min induction of *rpoH*. This operon encodes genes involved in redox reactions and acyl-CoA metabolism, and it forms a distinct cluster #5 (Appendix A). The gene cluster from PA1211-1221 (STRING cluster #1 in Appendix A) is also highly induced by increased RpoH (two to 80-fold induction). This operon is predicted to be involved in non-ribosomal peptide antibiotic biosynthesis. The operon includes genes with α/β hydrolase domains involved in non-ribosomal peptide synthesis (PA1211 and PA1219) as well as genes for Gamma-butyrobetaine dioxygenase (PA1213) and a S- adenosyl-methionine methyl transferase (PA1216). Since the gene cluster from PA1211-PA1221 was not shown to be induced by heat shock [25], we verified the transcriptomics results by measuring mRNA level by RT-qPCR of PA1216, the most highly induced gene based on the transcriptomics analysis. Following 15 min of arabinose induction of *P. aeruginosa* ∆*rpoH* (P_BAD_-*rpoH*), PA1216 had a nine-fold increase in mRNA (*p* < 0.0001) (Figure 2A). We also assayed mRNA increase in PA1216 following heat shock of wild-type *P. aeruginosa* PAO1 from 37 °C to 45 °C for 15 min by RT-qPCR. The increase in mRNA from heat shock (five-fold, *p* < 0.0001) (Figure 2B) indicates that this operon is under control of RpoH and is regulated by heat shock.

Another large gene cluster for cell wall and membrane synthesis and modification, PA5001-PA5010 (interconnected protein cluster #3 in Appendix A), was also significantly induced by increased RpoH levels (2.5 to 11.7-fold). These genes encode glycosyl transferases and genes required for biosynthesis of the lipopolysaccharide core (*waaP* and *waaG*) [35]. Many of these genes are classified as essential by Turner et al. [29] (Table 5). Since the PA5001-5010 gene cluster was not induced by heat shock in the previous heat shock studies, we analyzed the effect of heat shock and induction of RpoH on the mRNA levels of PA5008/*wapP,* by qRT-PCR. PA5008 increased seven-fold due to heat shock in wild-type *P. aeruginosa* PAO1 (*p* < 0.0001) and approximately 32-fold (*p* < 0.0001) following 15 min of arabinose induction in *P. aeruginosa* ∆*rpoH* (P_BAD_-*rpoH*) (Figure 2), indicating that this gene cluster is part of the RpoH regulon and is also involved in the heat shock response.

**Genes with reduced expression due to a sudden increase in RpoH.** We analyzed by STRING [33] the list of genes that had reduced mRNA abundance, to understand functional protein–protein interaction networks of genes with reduced expression due to increased RpoH (Appendix A and Appendix A). Cluster #3 in Appendix A contains several putative sigma factors, such as PA1911/*femR*, PA2468/*foxI* and PA1300/*hxuI*. Therefore, it is likely that some of the RpoH-induced expression changes may be due to indirect effects through these other effectors. Cluster #3 contains genes involved in heme uptake and transport, including several ferric and pyoverdine receptors. Cluster #2 is enriched in genes for pyoverdine synthesis and export, while cluster #4 has genes of the *fpvGHJKCDEF* operon that play a role in iron acquisition by pyoverdine [36]. Therefore, RpoH induction downregulates genes with a broad function in heme and iron uptake and metabolism. Cluster #1 among the downregulated genes (Appendix A) includes several components of the type VI secretion system, along with some poorly characterized genes.

**Overlap analysis of RpoH-induced genes.** We performed an overlap analysis of RpoH-induced genes with published gene lists to identify similarities between the RpoH regulon and other functional gene groups (Table 6) as in ref. [16]. Not surprisingly, the greatest overlap occurred between the RpoH regulon shown here, and the heat shock response in *P. aeruginosa* PAO1 [25] (www.pseudomonas.com, accessed on 2 February 2022). Other significant overlap occurred between genes induced by RpoH and genes highly expressed in a mouse wound biofilm infection [24]. The RpoH-induced genes here had significant overlap with genes associated with resistance to ciprofloxacin [37] and tobramycin [38], but not to genes associated with ciprofloxacin tolerance [37] (Table 6). There was no significant overlap with genes induced by ciprofloxacin or tobramycin in drip-flow biofilms [16]. The RpoH-regulated genes had significant overlap with genes induced by MvfR [39]. The downregulated gene set had the greatest overlap with genes for iron limitation [40,41], RpoS-regulated genes [42], stationary phase genes [43,44,45,46], and genes involved in virulence (www.pseudomonas.com), peroxide stress [47,48,49,50], and biofilms [51,52,53,54,55,56,57,58] (Table 6).

**RpoH levels modulate sensitivity to aminoglycoside antibiotics.** To construct the *rpoH* deletion, we used the gentamicin resistance gene [59]. During mutant construction, we noticed an increase in the sensitivity of the mutant strain to gentamicin. Similarly, a *Vibrio cholerae* ∆*rpoH* mutant had enhanced sensitivity to the aminoglycoside kanamycin [7] (*P. aeruginosa* PAO1 is naturally resistant to kanamycin). In addition, the analysis in Table 6 showed significant overlap between genes that were identified by Tn-seq mutants exhibiting hypersensitivity to tobramycin [38]. Therefore, we determined the effect of aminoglycosides and other classes of antibiotics on *P. aeruginosa* ∆*rpoH* (P_BAD_-*rpoH*). Using antibiotic disc diffusion assays, we compared wild-type *P. aeruginosa* PAO1 with *P. aeruginosa* ∆*rpoH* (P_BAD_-*rpoH*) cultured on cation-adjusted Mueller Hinton medium in the presence of 0.1% arabinose, 1% arabinose, or with no added arabinose. The results showed increased sensitivity (increased zone of clearing) of *P. aeruginosa* ∆*rpoH* (P_BAD_-*rpoH*) to each of three aminoglycosides (kanamycin, gentamicin, and tobramycin) compared to *P. aeruginosa* PAO1 (*p* < 0.0001) (Figure 3). As controls to determine if the plasmids used in these studies affected sensitivity to these antibiotics, we tested wild-type *P. aeruginosa* PAO1 (pCF430) and *P. aeruginosa* PAO1 (pMF477) by the disc diffusion assay. Neither plasmid influenced the zone of clearing of *P. aeruginosa* PAO1 to any of these antibiotics (Appendix A). *P. aeruginosa* PAO1 showed no zone of clearing in the disk diffusion assay with kanamycin, while *P. aeruginosa* ∆*rpoH* (P_BAD_-*rpoH*) had a zone of clearing around the kanamycin disk. The zone size for kanamycin was significantly reduced when 1% arabinose was added to the medium (*p* = 0.0023), but not to the level of the wild-type strain (Figure 3A). *P. aeruginosa* ∆*rpoH* (P_BAD_-*rpoH*) also had significantly increased zones of clearing compared to the wild-type strain for both gentamicin and for tobramycin (*p* < 0.0001) (Figure 3B,C). Arabinose addition to the medium caused a decrease in the zone size, but not to wild-type levels. The *P. aeruginosa* ∆*rpoH* (P_BAD_-*rpoH*) strain did not have an increased zone of clearing compared to the wild-type strain for other antibiotics, including chloramphenicol, trimethoprim, nalidixic acid, cephalothin, and clindamycin. We also tested the strain with a Tn5 insertion in the 5′ end of the *rpoH* gene, obtained from the *P. aeruginosa* two-allele library (*P. aeruginosa rpoH*::Tn5-39115) for sensitivity to aminoglycoside antibiotics. The *P. aeruginosa rpoH*::Tn5-39115 strain gave zones of clearing to all antibiotics tested that were not significantly different from the wild-type strain (*p* > 0.05) (Figure 3).

Given that *rpoH* is an essential gene and that *P. aeruginosa rpoH*::Tn5-39115 gave similar antibiotic resistance patterns to the wild-type strain, the transposon in this strain likely does not disrupt RpoH activity.

We next characterized the sensitivity of *P. aeruginosa* ∆*rpoH* (P_BAD_-*rpoH*) to aminoglycosides using minimum inhibitory concentration (MIC) of planktonic cells cultured in the wells of microtiter plates. We tested two aminoglycosides, gentamicin and tobramycin, and a fluoroquinolone, ciprofloxacin. The results showed increased sensitivity of *P. aeruginosa* ∆*rpoH* (P_BAD_-*rpoH*) to gentamycin and tobramycin compared to *P. aeruginosa* PAO1, but no difference when exposed to ciprofloxacin (Figure 4). As with the disk diffusion assay, the plasmids pCF430 and pMF477 in wild-type *P. aeruginosa* PAO1 did not cause an increased sensitivity to gentamicin or tobramycin (Appendix A).

**Effect of tobramycin and ciprofloxacin on *P. aeruginosa* ∆*rpoH* (P_BAD_-*rpoH*) cultured in biofilms.** To determine if RpoH affects antibiotic tolerance of *P. aeruginosa* cultured in biofilms, we cultivated wild-type *P. aeruginosa* PAO1 and *P. aeruginosa* ∆*rpoH* (P_BAD_-*rpoH*) in colony biofilms for three days in the presence or absence of 1% arabinose. Colony biofilms were transferred to fresh medium every 24 h. Following three days of growth, biofilms were transferred to medium containing 10 μg × mL^−1^ tobramycin (aminoglycoside antibiotic) or 1 μg × mL^−1^ ciprofloxacin (fluoroquinolone antibiotic). These antibiotic concentrations are approximately 20× the MIC of tobramycin for *P. aeruginosa* PAO1 growing in planktonic culture and approximately 16× the MIC to ciprofloxacin for *P. aeruginosa* PAO1 growing in planktonic culture. Colony biofilms were transferred to fresh medium containing antibiotics every 24 h. The number of viable cells that survived 1, 2, and 3 days of antibiotic exposure was determined as colony forming units (CFUs), and the log reduction in viable cells was determined by comparing CFUs after treatment to those prior to antibiotic treatment. Four biological replicates were performed per antibiotic. The no-antibiotic control biofilms showed no loss of viability over the course of six days of biofilm growth (Figure 5A). For the ciprofloxacin-treated biofilms, the wild-type strain had approximately 1.1–1.5 log reduction of viable cells over the three days of ciprofloxacin treatment. In the absence of arabinose addition, the *P. aeruginosa* ∆*rpoH* (P_BAD_-*rpoH*) strain had a 1.7–1.8 log reduction in viable cells over the three-day course of treatment (Figure 5B). Addition of arabinose to the medium for strain *P. aeruginosa* ∆*rpoH* (P_BAD_-*rpoH*) resulted in decreased sensitivity (1.2–1.4 log reduction) of viable cells during three days of ciprofloxacin treatment. However, the loss of viability for the mutant strain, both in the presence and absence of arabinose, was not statistically significantly different from the wild-type strain at any of the time points (*p* > 0.1). For the tobramycin treatment, the *P. aeruginosa* ∆*rpoH* (P_BAD_-*rpoH*) strain did not show a statistically significant difference from the wild-type strain after one day of tobramycin treatment (*p* = 0.68) (Figure 5C). However, after two and three days of tobramycin treatment, the *P. aeruginosa* ∆*rpoH* (P_BAD_-*rpoH*) strain had greater loss of viability compared to the wild-type strain (*p* < 0.0001 and *p* < 0.001, respectively). Addition of arabinose to the medium for the *P. aeruginosa* ∆*rpoH* (P_BAD_-*rpoH*) strain restored the loss of viability to levels that were not statistically different from the wild-type strain (*p* > 0.06).

**RpoH affects survival of *P. aeruginosa* during starvation.** To determine the effect of RpoH on survival of *P. aeruginosa* during starvation, we cultured *P. aeruginosa* PAO1 and *P. aeruginosa* ∆*rpoH* (P_BAD_-*rpoH*) to early stationary phase in the presence and absence of arabinose. The cells were washed, resuspended in phosphate-buffered saline (PBS) with or without arabinose, and incubated for up to 5 days at 37 °C, with shaking. The cultures were sampled daily and assayed for viable cells as colony forming units (CFU). Here, as in our previous study [60], wild type *P. aeruginosa* PAO1 survived starvation with no loss of viability for 5 days (*p* > 0.99) (Figure 6A). In contrast, the *P. aeruginosa* ∆*rpoH* (P_BAD_-*rpoH*) strain had a significant loss in viability (*p* < 0.0001), with a greater than 2-log reduction of viable cells after 3 days of starvation (Figure 6A). Viability of *P. aeruginosa* ∆*rpoH* (P_BAD_-*rpoH*) during starvation was not restored when arabinose was added to the growth medium and the starvation medium (*p* < 0.0001). There was no effect on survival during starvation due to the wild-type strain *P. aeruginosa* PAO1 carrying plasmid pCF430 or pMF477 (Appendix A).

We performed a similar starvation survival experiment on wild-type *P. aeruginosa* and *P. aeruginosa* ∆*rpoH* (P_BAD_-*rpoH*) cultured in biofilms. Biofilms were cultured on TSA or TSA with 0.1% arabinose with transfers to fresh plates daily for two days. The colony biofilms were then transferred to PBS agar plates containing 0 or 0.1% arabinose, with transfers to fresh medium every 24 h for up to 5 days. Starved biofilms were assayed daily for viability as CFUs. The results showed no significant change in viability for the *P. aeruginosa* ∆*rpoH* (P_BAD_-*rpoH*) strain or the wild-type strain over five days of starvation (*p* > 0.23 for all strains and conditions) (Figure 6B).

## 3. Discussion

During the heat-shock response, the cellular concentration of the heat-shock sigma factor RpoH (σ^32^) temporarily increases as it becomes released from molecular chaperones and as its expression increases. A primary function of RpoH is to regulate expression of genes that are involved in protein repair and degradation. Here, we show that a sudden increase in RpoH in *P. aeruginosa* results in an increase in the mRNA abundances for 29 proteases, chaperones, and protease/chaperone accessory proteins. Expression of many of these genes was also shown to be induced by heat-shock [25]. Ten of these proteases and chaperones are essential for viability of *P. aeruginosa* during growth under laboratory conditions. The essential proteases and chaperones, such as ClpX, Lon, DnaJ, DnaK, CarB, GrpE, and FtsH, provide housekeeping functions for protein processing and protein folding, and therefore could be considered “housekeeping proteins” even though they are regulated by the alternative σ^32^ sigma factor. Therefore, the RpoH regulon, usually considered a response to an environmental stress, includes a subset of genes for housekeeping functions. Some of these housekeeping genes are over-expressed when the cells experience conditions that cause proteins to be misfolded, allowing the cells to survive environmental stresses. Since the RpoH sigma factor regulates proteins that are essential for *P. aeruginosa*, regulation of one or more of these essential proteins is likely the reason that RpoH is an essential sigma factor in *P. aeruginosa* and why the *rpoH* gene cannot be deleted unless another copy of *rpoH* is in the cells.

In *P. aeruginosa*, expression of the *rpoH* gene is controlled by three promoters, including two housekeeping promoters with consensus sequences for the σ^D^ binding site (P1 and P2) and a third promoter (P3) that is regulated by the stress response sigma factor σ^E^ [61]. σ^E^ also regulates the conversion to the mucoid phenotype (alginate production) found in *P. aeruginosa* isolates from pulmonary tissue of chronically infected cystic fibrosis patients [62,63]. In mucoid stains of *P. aeruginosa*, the σ^E^ sigma factor is released from its molecular chaperones *mucA* and *mucB* [64], causing increased expression from the *algD* promoter, and of the alginate biosynthetic operon. Here, we observed a 2.1- and 2.9-fold and significantly different (*p* = 0.0025 and *p* = 0.0019, respectively) increased expression of *mucA* and *mucB* following 15 min of an increase in the cellular concentration of RpoH, further showing an interrelation between the σ^E^ and σ^32^ stress-response sigma factors.

In *P. aeruginosa*, increased RpoH caused increased expression of other genes that are required for stress response. These include genes involved in nucleic acid metabolism and recycling (Table 3). Genes involved in DNA metabolism possibly allow repair of damaged DNA. These genes encode proteins for the DNA helicases TopA (topoisomerase) and PA3272 (DNA helicase). TopA and the nucleic acid metabolism protein CpdA (3′,5′-cyclic-AMP phosphodiesterase) are essential for *P. aeruginosa* under laboratory growth conditions [29] and are upregulated due to a sudden increase in RpoH. TopA was also shown to be upregulated by heat shock [25]. Another nucleic acid metabolism protein with increased mRNA abundance due to an increase in RpoH is the RNA binding protein Hfq. Hfq is involved in mRNA folding and binding of small regulatory RNAs. In *P. aeruginosa*, Hfq has a putative upstream σ^32^ promoter consensus sequence, suggesting a link between the heat-shock response and RNA:RNA folding.

Other genes with housekeeping functions that increase due to an increase in σ^32^ levels are genes involved in respiration, including several cytochrome genes and the *nuo* operon. The *nuo* operon encodes the large membrane associated NADH dehydrogenase complex and is involved in oxidative phosphorylation. The *nuo* operon is essential for *P. aeruginosa* anaerobic respiration with nitrate as the terminal electron acceptor [65,66], and in *E. coli* provides a competitive advantage for the cells during stationary phase [67]. Interestingly, mutations in the *nuo* operon increase aminoglycoside tolerance, possibly by downregulating the energy metabolism required for aminoglycoside uptake [68,69].

Expression of many genes not involved in protein degradation and repair are induced due to increased RpoH levels. Of particular interest are the highly expressed gene clusters from PA0744-PA0747, PA1221-PA1211, and PA5001-PA5010. The PA0744 operon includes enzymes for fatty acid biosynthesis, including Enoyl-CoA hydratase/isomerase (PA0744 and PA0745), Acyl-CoA dehydrogenase (PA0746), and NAD-dependent aldehyde dehydrogenase (PA0747). The structure and function of PA0745 (DspI) were determined [70] and shown to be involved in the dehydration reaction cis-2-decenoic acid, a signaling compound involved in biofilm dispersion [71]. A gene cluster from PA1221-1211 (possibly consisting of two operons) is significantly upregulated by a 15 min increase in RpoH (two- to 80-fold increase). The genes encode for proteins for non-ribosomal peptide synthesis (PA1221) and a condensation domain (PA1220) involved in production of peptide antibiotics. The gene cluster also includes a protein with an S-adenosyl-L-methionine-dependent methyltransferase domain (PA1216) that was upregulated 80-fold following induction with RpoH. Using a random transposon screening assay, PA1216 was identified as necessary for *P. aeruginosa* virulence in a *C. elegans* infection model [72]. The PA5001-PA5010 operon is significantly upregulated by a 15 min increase in RpoH levels (2.5- to 12-fold). The PA5001-PA5010 operon includes the lipopolysaccharide kinase *waaP*, that modifies the inner core heptose of lipopolysaccharide with phosphate groups. Mutations in *waaP* affect outer membrane permeability due to aberrant lipopolysaccharide biosynthesis, that lacks core sugars and phosphates, and that fail to reach the outer membrane [35]. As a result, expression of *waaP* affects the sensitivity of *P. aeruginosa* to certain antibiotics, including novobiocin, and to sodium dodecyl sulfate [73]. We verified induced expression of *waaP* due to increased RpoH levels and due to heat shock by RT-qPCR. The induced expression of the PA5001-PA5010 gene cluster by an increase in the cellular concentration of RpoH again shows a link for RpoH regulation of essential housekeeping functions in *P. aeruginosa*.

**Effect of RpoH on long-term survival of *P. aeruginosa*.** In other studies, we characterized proteins required for long-term survival of *P. aeruginosa* [60,74,75]. In those studies, we showed that the ribosome hibernation promoting factor (HPF) is required for optimal recovery of *P. aeruginosa* from starvation, by protecting ribosomes from degradation when the cells are starved. Here, we show that RpoH is also required for optimal recovery of *P. aeruginosa* from starvation, even though RpoH does not regulate expression of *hpf* [76]. The molecular mechanism for RpoH protection of cell viability during starvation is difficult to discern, compared to the mechanism for HPF. Rather than a single gene deletion, RpoH is a global regulator of genes required for both stress response and for housekeeping functions. Therefore, it is difficult to determine which of these RpoH-regulated genes is essential for long-term survival during starvation. The *P. aeruginosa* ∆*rpoH* (P_BAD_-*rpoH*) strain still contains leaky expression of *rpoH*, even when not induced. Rather than comparing a deletion mutant strain with the wild-type strain, we were comparing a strain with disrupted *rpoH* expression to the wild-type strain [77]. However, the *P. aeruginosa* ∆*rpoH* (P_BAD_-*rpoH*) strain had a significant loss in viability (greater than two log reduction) (*p* < 0.0001), while there was no loss of viability of the wild-type strain under similar conditions (*p* > 0.99). Addition of arabinose to *P. aeruginosa* ∆*rpoH* (P_BAD_-*rpoH*) during growth and during starvation did not complement the starvation survival defect. Therefore, the starvation survival phenotype observed for *P. aeruginosa* ∆*rpoH* (P_BAD_-*rpoH*) strain is not solely due to the cellular concentration of RpoH, but rather to the disruption of RpoH regulation, which may occur at the transcriptional, post-transcriptional, and post-translational level.

Based on the transcriptomic data of the RpoH regulon, it is possible to speculate on mechanisms for reduced survival of the *P. aeruginosa* ∆*rpoH* (P_BAD_-*rpoH*) strain during starvation. Mechanisms could include improper processing of misfolded proteins. Since RpoH controls the expression of 29 proteases, chaperones, and accessory proteins, disruption of RpoH could result in build-up of toxic misfolded proteins while the cells are starving. Linder et al. [22,78] described the effect of the build-up of misfolded proteins on cell aging and showed that misfolded proteins are translocated to the cell poles by the molecular chaperone IbpA in the aging subpopulation of cells. IbpA is highly induced by a sudden increase in RpoH levels and by heat shock. Another possible mechanism for reduced survival of *P. aeruginosa* ∆*rpoH* (P_BAD_-*rpoH*) during starvation is the disruption of membrane potential of the starved cells. Many genes, including cytochromes, NADH dehydrogenase, and genes required for maintenance of membrane potential are controlled by RpoH. Cells must maintain a membrane potential in order to recover from starvation [79]. Disruption of expression of these genes involved in oxidative phosphorylation may influence the ability of *P. aeruginosa* to survive starvation. RpoH also affects expression of the PA5001-PA5010 operon, which includes several essential enzymes involved in lipopolysaccharide modification. Disruption of LPS structure could impair membrane integrity and the ability of *P. aeruginosa* to recover from extended starvation. Since RpoH also controls expression of many essential proteins, it is also possible that low level expression of essential genes is required for the cells to survive during low nutrient conditions and to recover from starvation.

The loss of viability of the *P. aeruginosa* ∆*rpoH* (P_BAD_-*rpoH*) strain during starvation was much greater when the bacteria were starved planktonically than when they were starved in biofilms. Although genes for the RpoH regulon, such as *ibpA*, are highly expressed during biofilm growth [21], the *P. aeruginosa* ∆*rpoH* (P_BAD_-*rpoH*) biofilms were resilient to starvation conditions when they were starved as intact biofilms.

**Effect of RpoH on aminoglycoside antibiotic resistance of *P. aeruginosa*.** The *P. aeruginosa* ∆*rpoH* (P_BAD_-*rpoH*) strain, in addition to being impaired in survival during starvation, is also more sensitive to aminoglycoside antibiotics, when compared to the wild-type strain. The sensitivity of the *P. aeruginosa* ∆*rpoH* (P_BAD_-*rpoH*) strain is specific for aminoglycosides (kanamycin, gentamicin, and tobramycin) and not for other classes of antibiotics, such as the fluoroquinolone, ciprofloxacin. Since aminoglycosides inhibit the ribosome and protein synthesis, a likely mechanism for the increased sensitivity to aminoglycosides is improper processing of misfolded proteins, due to the misregulation of proteases and chaperones. AsrA, an ATP-dependent protease which is regulated by RpoH, is required for short-term protection against lethal levels of tobramycin and may also play a role in the enhanced sensitivity of the mutant strain to aminoglycosides [27]. The addition of arabinose to the growth medium partially restored the level of resistance of *P. aeruginosa* ∆*rpoH* (P_BAD_-*rpoH*) to aminoglycosides. However, addition of arabinose did not restore planktonic MICs to wild-type levels, suggesting that disruption of the proper regulation of RpoH, rather than the cellular concentration of RpoH, is responsible for the phenotypic effect on aminoglycoside tolerance. The increased sensitivity of *P. aeruginosa* ∆*rpoH* (P_BAD_-*rpoH*) biofilms to tobramycin was statistically significant compared to the wild-type strain. Although the *P. aeruginosa* ∆*rpoH* (P_BAD_-*rpoH*) biofilms were more sensitive to tobramycin compared to the wild-type strain, there was only a 1.2 log reduction in viability of the mutant strain after three days of continuous antibiotic treatment with 20× the MIC of tobramycin. The *P. aeruginosa* ∆*rpoH* (P_BAD_-*rpoH*) strain had a 1.8 log reduction in viability after three days of treatment with 16× the MIC for ciprofloxacin. The biofilm results further demonstrate the resilience of *P. aeruginosa* growing in biofilms to antibiotics, even with disruption of a major sigma factor involved in essential cellular functions and stress response.

## 4. Materials and Methods

**Bacterial strains, plasmids, and media**. *Escherichia coli* and *P. aeruginosa* were routinely cultured in LB Miller broth (10 g tryptone, 5 g yeast extract, and 5 g NaCl × L^−1^). *Pseudomonas* isolation agar (Difco Laboratories, Franklin Lakes, NJ, USA) was used to select for *P. aeruginosa* after conjugation with *E. coli*. Antibiotics, when used, were at the following concentrations: ampicillin at 100 µg × mL^−1^, carbenicillin at 150 µg × mL^−1^, and tetracycline at 40 µg × mL^−1^. For *rpoH* induction experiments, *P. aeruginosa* ∆*rpoH* (P*_BAD_*-*rpoH*) was cultured in Biofilm Minimal Medium (BMM) [80,81]. BMM consisted of 9.0 mM sodium glutamate, 50 mM glycerol, 2 mM MgSO_4_, 0.15 mM NaH_2_PO_4_, 0.34 mM K_2_HPO_4_, 145 mM NaCl, 2 mM CaCl_2_ · 2H_2_O, 20 μL trace metals, and 1 mL vitamin solution per liter. The pH of the medium was adjusted to 7.0. The trace-metal solution contained 5.0 g CuSO_4_ · 5H_2_O, 5.0 g ZnSO_4_ · 7H_2_O, 5.0 g FeSO_4_ · 7H_2_O, and 2.0 g MnCl_2_ · 4H_2_O per liter of 0.83 M HCl. The vitamin solution contained 0.5 g thiamine and 1 mg biotin per liter. All chemicals were obtained from Fisher Scientific (Waltham, MA, USA) or Sigma-Aldrich (St. Louis, MO, USA).

To construct a strain of *P. aeruginosa* PAO1 with *rpoH* controlled by the arabinose inducible P_BAD_ promoter, the *rpoH* gene was amplified from *P. aeruginosa* genomic DNA using primers RpoH Nhe 5′ and RpoH Hind 3′ (Appendix A). The PCR product was ligated into the *Nhe*I and *Hind*III sites of the low-copy number IncP plasmid pCF430, which contains the P_BAD_ promoter and the AraC repressor (generously provided by Dr. Clay Fuqua), forming plasmid pMF477. The *rpoH* gene was deleted from *P. aeruginosa* PAO1(pMF477) using allelic exchange. For allelic exchange, DNA upstream of *rpoH* was amplified using primers RpoH *Hind*III 5′ Up and RpoH T7 Up R Acc (Appendix A). Downstream DNA from *rpoH* was amplified using primers RpoH *Hind*III 3′ Down and RpoH T7 Down F Acc (Appendix A). The two PCR products were spliced together using overlap extension PCR, then ligated into plasmid pEX18T using the *Hind*III restriction site. The gentamicin resistance cassette, containing flanking *frt* sites, was ligated between the upstream and downstream DNA fragments using the Acc65I restriction site, producing plasmid pMF431. Plasmid pMF431 was used to replace *rpoH* on the genome of *P. aeruginosa* with the gentamicin resistance gene. The gentamicin resistance gene was removed from the genome using FLP/FRT recombination as described previously [59]. Restriction endonucleases, DNA ligase, and Taq polymerase were purchased from New England Biolabs (Ipswich, MA, USA). The *rpoH* transposon mutant was obtained from the *P. aeruginosa* two-allele transposon library [82].

**Growth and treatment of cultures for heat and *rpoH* induction.** Strains were cultivated overnight at 37 °C in Tryptic Soy Broth amended with 40 µg × mL^−1^ tetracycline and 0.1% arabinose for the PAO1∆*rpoH* (P_BAD_-*rpoH*) strain. Two 1 mL aliquots of the overnight cultures were washed twice with 1 mL phosphate-buffered saline (PBS) pH 7.0 (7.7k RCF, 1 min) and resuspended in 1 mL PBS. Each washed aliquot of the overnight culture was added to 25 mL of Biofilm Minimal Medium (BMM) in a 125 mL baffled flask and incubated with shaking at 200 rpm at 37 °C for 7.75 h.

For heat treatment, following 7.75 h of incubation in BMM, two 4 mL aliquots were removed from each culture. The control aliquot was kept at 37 °C, while the other aliquot was transferred to a 45 °C water bath to induce heat shock for 15 min. Aliquots (2 mL) were removed, and cells were collected by centrifugation at 7.7k RCF for 2 min at 4 °C. The cell pellets were frozen at −80 °C until RNA extraction and analysis. Three independent biological replicates were performed for each strain and condition.

For *rpoH* induction in *P. aeruginosa* PAO1∆*rpoH* (P_BAD_-*rpoH*), following 7.75 h of growth in BMM, arabinose was added to a final concentration of 0.1% in one of the two flasks. After 15 min, cells were collected from the non-treated control and arabinose treated cultures by centrifugation at 7.7k RCF for 2 min at 4 °C. The cell pellets were frozen at −80 °C for RNA extraction. Six biological replicates were performed for each strain and condition, with three replicates used for microarray analysis and three replicates used for RT-qPCR analysis.

**RNA extraction.** RNA was isolated from cells using the hot phenol extraction method with chemicals obtained from Fisher Scientific (Waltham, MA, USA). Briefly, pelleted cells were resuspended in 200 µL lysis buffer (0.15 M sucrose, 0.01 M sodium acetate, pH 4.5) and 200 µL 2% sodium dodecyl sulfate. Following the addition of 400 µL phenol, the mixture was incubated for 5 min at 65 °C with frequent vortexing. A Heavy Phase Lock Gel (PLG) tube (5 Prime Therapeutics, South San Francisco, CA, USA) was used according to the manufacturer’s instructions to separate the phases. A second cleanup was performed in the PLG tube with 400 µL phenol:chloroform:isoamyl alcohol (25:24:1) solution. The aqueous phase was removed, and the RNA was purified using the Direct-zol RNA MiniPrep kit (Zymo Research, Irvine, CA, USA) with an on-column DNase treatment according to the manufacturer’s instructions. The purified RNA was then treated with turbo-DNase using the Turbo-DNA free kit (Invitrogen, now Thermo Fisher Scientific, Waltham, MA, USA) following the manufacturer’s instructions. RNA quality was assessed on the Bioanalyzer 2100 using an RNA6000 nano assay (Agilent Technologies, Santa Clara, CA, USA), and all samples were found to be intact with RNA Integrity Number (RIN) scores > 9.

**RT-qPCR.** One-step reverse transcription-quantitative PCR (RT-qPCR) was performed with the Rotor-Gene SYBR green RT-PCR kit (Qiagen, Germantown, MD, USA) as described previously [21,60]. Three biological replicates for each strain and condition were assayed in triplicate using primers designed for the *acpP*, *rpoH*, *ibpA*, *groES*, *wapP*, *hpf*, and PA1216 transcripts (Appendix A) obtained from IDT Technologies (Coralville, IA, USA). Primer sets were validated by determining efficiencies obtained from 10-fold serial dilution curves of *P. aeruginosa* RNA, using the equation *E* = 10^−1/slope^ − 1. RT-qPCR efficiencies were similar for all primer sets*,* with *r*^2^ > 0.98. One-step RT-qPCR reactions were assembled according to the Rotor-Gene SYBR green RT-PCR handbook (Qiagen, Germantown, MD) using 2 μL of template RNA at ~10 ng x µL^-1^ and 250 nM of each primer and carried out in a Rotor-Gene 6000 instrument (Qiagen, Germantown, MD, USA). Cycling parameters were as follows: one cycle of 55 °C for 12 min, denaturation at 95 °C for 6 min, and then 50 cycles of 95 °C for 5 s and 60 °C for 12 s. Data were acquired during the 60 °C annealing step, and a melt curve was generated at the conclusion of each run. Negative controls lacking reverse transcriptase were performed with each RNA sample and revealed that samples were free from DNA contamination. Mean fold change (+/− standard deviation) of each transcript in response to arabinose or heat treatment was calculated using the formula: fold change = 2^−ΔΔ*C*^_T_ [83] using the constitutively expressed housekeeping gene *acpP* as the normalization control gene. Three independent biological replicates were performed, and the fold change of each sample was plotted using GraphPad Prism version 8.3.0. Mean fold change was plotted as a dashed line. To determine if transcripts were significantly upregulated by heat or by arabinose, *t*-tests using the Holm–Sidak correction test were performed using GraphPad Prism version 8.3.0 at *alpha* = 0.01.

**Transcriptome analysis.** Samples were prepared for microarrays as described previously [16,17]. Briefly, 8 μg of Turbo DNase-treated RNA was reverse transcribed, fragmented, and biotin end labeled according to the Affymetrix prokaryotic target labeling protocol (GeneChip expression analysis technical manual, November 2004). Labeled cDNA was then hybridized to Affymetrix *P. aeruginosa* microarrays (part 900339) for 16 h at 50 °C with constant rotation. Microarrays were stained using a GCOS Fluidics Station 450 and scanned with an Affymetrix 7G scanner. Affymetrix GCOS v1.4 was used to generate CEL files, which were imported into FlexArray v1.6.1 [84] for quality control, normalization, and data analysis.

Microarray data from three biological replicates of 8 h planktonic arabinose induced PAO1∆*rpoH* (P_BAD_-*rpoH*) cultures and three non-induced PAO1∆*rpoH* (P_BAD_-*rpoH*) cultures were background corrected and normalized using the GC-RMA algorithm in FlexArray v1.6.1. Genes with statistically significant changes in expression (>twofold change and *p* < 0.05) as determined by analysis of variance (ANOVA) were uploaded into the Database for Annotation, Visualization, and Integrated Discovery (DAVID) [85,86]. The Functional Annotation Clustering Tool was used to identify enriched biological processes in the study. The most significant term in each cluster was selected as the category identifier. Each *p* value is the geometric mean of all the enrichment *p* values (Expression Analysis Systematic Explorer [EASE] scores) of each annotation term in the group, calculated by the formula *p* = 10^–EASE^. The EASE score was determined in DAVID with a modified Fisher exact test (log-transformed average *p* value). To identify the physiological activities represented by the lists of significantly up- and down-regulated genes, the lists were compared to lists of genes associated with responses or activities, such as iron limitation, quorum sensing, or antibiotic stress, described in reference [16]. *P* values for assessing the statistical significance of gene set enrichment were calculated using a negative binomial distribution [16,17]. Microarray data have been deposited in the NCBI Gene Expression Omnibus [87] and are accessible through GEO series accession number GSE217157.

**Antibiotic susceptibility assays.** Antibiotic susceptibility assays were performed using the disc diffusion assay. Discs containing 10 µg tobramycin, 10 µg gentamicin, 30 µg kanamycin, 30 µg nalidixic acid, 30 µg cephalothin, 2 µg clindamycin, 25 µg trimethoprim, and 30 μg chloramphenicol were obtained from Hardy Diagnostics. *P. aeruginosa* PAO1, *P. aeruginosa rpoH*::Tn5—39115, *P. aeruginosa* ∆*rpoH* (P_BAD_-*rpoH*), *P. aeruginosa* PAO1 (pCF430), and *P. aeruginosa* PAO1 (pMF477) strains were cultured in Mueller Hinton II Broth until reaching an optical density equivalent to a 0.5 McFarland Standard. The cultures were spread plated using a cotton swab onto Mueller Hinton II agar plates with or without added arabinose to allow confluent growth. Antibiotic discs were placed onto the inoculated plates and incubated for 16 h at 37 °C. The zone of inhibition was measured and recorded. A minimum of three biological replicates was tested. The mean and standard deviation of the collected measurements were plotted using GraphPad Prism version 8.3.0. To determine if the antibiotic susceptibility in the mutated strains was significantly different from PAO1, a two-way ANOVA was performed with Holm–Sidak correction for multiple testing to generate adjusted *p*-values at *alpha* = 0.01 using GraphPad Prism version 8.3.0.

Antibiotic susceptibility assays were also performed using the broth microdilution assay to determine the Minimum Inhibitory Concentration (MIC) [17,88] of tobramycin, gentamicin, and ciprofloxacin with 0.1% and no added arabinose. For MIC assays, PAO1 and PAO1∆*rpoH* (P_BAD_-*rpoH*)*,* PAO1 (pCF430), and PAO1 (pMF477) were cultured overnight in cation adjusted Mueller Hinton Broth containing 10 mg MgCl_2_ and 20 mg CaCl_2_ per liter (MHII Broth), amended with 40 ug × mL^−1^ tetracycline for growth of the mutant strain. Cultures were diluted 10-fold into fresh MHII Broth, and the optical density at 600 nm (OD_600_) was determined using a Cecil CE2041 Spectrophotometer. Cultures were then diluted to an OD_600_ of 0.1 and placed in the wells of clear, sterile, flat-bottom 96-well microtiter plates (CytoOne, USA Scientific, Ocala, FL, USA). Two-fold serial dilutions of antibiotics were also added to the wells of the microtiter plates. Growth control wells containing no antibiotics and sterility control wells containing media but lacking bacterial cells were included on every run. The microtiter plates were placed in an Epoch 2 plate reader (BioTek, Winooski, VT, USA) and incubated at 37 °C with shaking, and the OD_600_ was assayed every 15 min for 20 h. The minimum antibiotic concentration that prevented growth at 1.0 OD_600_ for 16 h was defined as the MIC. Three independent assays, which yielded identical MIC values, were performed. Results were plotted using GraphPad Prism version 8.3.0.

Antibiotic susceptibility assays were also performed on colony biofilms [21]. For these experiments, overnight *P. aeruginosa* PAO1 and PAO1 ∆*rpoH* (P_BAD_-*rpoH*) cultures grown in tryptic soy broth (TSB) were diluted into fresh TSB to an optical density at 600 nm (OD_600_) of 0.05, and 25-μL aliquots were used to inoculate sterile, 0.2-μm pore size, 13 mm diameter polycarbonate Isopore^TM^ membranes (Millipore, Burlington, MA, USA) positioned on TSA plates containing 1% arabinose or no added arabinose. Plates were inverted and incubated at 37 °C. Membranes containing colony biofilms were aseptically transferred to fresh TSA plates every 24 h for three days. Following three days of growth on TSA or TSA containing 1% arabinose, colony biofilms were transferred to TSA containing either 10 μg × mL^−1^ tobramycin sulfate, 1 μg × mL^−1^ ciprofloxacin hydrochloride, or no added antibiotic, and incubated at 37 °C for up to three additional days, with aseptic transfers to fresh plates every 24 h. Colony biofilms were disaggregated and viable cells were enumerated as CFU on TSA plates containing 1% or no added arabinose every 24 h for the duration of the study. Four independent biological replicates were performed. Log reduction in viability was calculated relative to the onset of treatment and plotted using GraphPad Prism version 8.3.0. To determine if the antibiotic susceptibility in the mutated strains was significantly different from that of PAO1, a two-way ANOVA was performed with the Holm–Sidak test to correct for multiple testing to generate adjusted *p*-values at *alpha* = 0.01 using GraphPad Prism version 8.3.0.

**Nutrient starvation assays.** Strains were cultured overnight in 3 mL TSB with 0.1% arabinose and without arabinose, amended with 40 µg × mL^−1^ tetracycline for strains containing plasmids pMF477 and pCF430. 120 μL of the overnight cultures was used to inoculate 4 mL of TSB with 0.1% and no added arabinose. The cultures were incubated at 37 °C with aeration for 6 h and then used for planktonic and biofilm starvation assays. Three independent biological replicates of each assay were performed.

For planktonic starvation assays, cell aliquots equaling 1 mL of 6.6 OD_600_ (Ultrospec Pro 2100, Amersham Biosciences, now GE Healthcare, Chicago, IL, USA) were washed twice in Phosphate Buffered Saline (PBS) pH 7.1 and then resuspended in 1 mL PBS. The washed cells were added to 25 mL PBS in 125 mL baffled flasks, incubated at 37 °C with shaking at 200 rpm, and assayed daily for viability as colony forming units (CFUs) using the drop plate method. The mean and standard error of the mean were calculated and plotted using GraphPad Prism v 8.3.0. Log reduction in viability was calculated relative to the onset of starvation and a two-way ANOVA was performed with the Holm–Sidak test to correct for multiple testing to generate adjusted *p*-values at *alpha* = 0.01 using GraphPad Prism version 8.3.0.

For biofilm starvation assays, cell aliquots were adjusted to 1 mL of 5.0 OD_600_ with the Ultrospec Pro 2100 (Amersham Biosciences, now GE Healthcare, Chicago, IL, USA), and 30 µL was used to inoculate sterile 0.2-μm pore size polycarbonate 13 mm diameter filters (Isopore^TM^ #GTTP01300, Millipore Sigma, Burlington, MA, USA) positioned on TSA containing 0.1% or no added arabinose. Agar plates were incubated at 37 °C and membranes containing colony biofilms were aseptically transferred to fresh TSA plates with 0.1% or no added arabinose after 24 h. After two days of growth on TSA or TSA with 0.1% arabinose, membranes containing colony biofilms were transferred to 1.5% agar plates containing phosphate-buffered saline pH 7.1 (PBSA) and 0 or 0.1% arabinose. Colony biofilms were incubated at 37 °C and transferred to fresh PBSA every 24 h for 5 days. Biofilms were assayed daily for viability as CFUs using the drop plate method. The mean and standard error of the mean were calculated and plotted using GraphPad Prism v 8.3.0.

**Bioinformatics analyses.** We used STRING network analysis [33] to visualize functional connections between genes. Two lists of significantly upregulated and downregulated genes were independently processed by STRING. The resulting networks were further modified to remove genes with no connections and those that had single links. Proteins were annotated using custom-made hidden Markov models (HMMs) for all PAO1 proteins. Briefly, for each protein, an automated search for homologs was done using HHblits [89]; the resulting hits were aligned and converted to HMMs. Both public and custom databases of protein families (e.g., Pfam; [90]) were interrogated using these HMMs. To further refine functional predictions, we queried AlphaFold2 models that have recently become available for many model organisms, including all PAO1 proteins [34]. Structural similarity to known proteins was established with Foldseek [91].

## Figures and Tables

**Figure 1 ijms-24-01513-f001:**
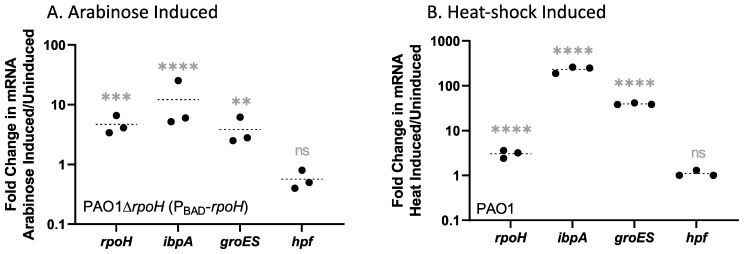
(**A**) Fold change of mRNA following 15 min of arabinose induction of *P. aeruginosa* ∆*rpoH* (P_BAD_-*rpoH*), showing significant difference of mRNA levels before and after arabinose treatment for *rpoH*, *ibpA*, and *groES*. (**B**) Fold change of *rpoH*, *ibpA*, and *groES* in wild-type strain *P. aeruginosa* PAO1, following 15 min of heat shock from 37 °C to 45 °C. No significant change was observed for the negative control mRNA *hpf* after either treatment. Three independent biological replicates were performed and the housekeeping gene *acpP* was used as an internal control. A two-way ANOVA was performed on delta Ct values pre- and post-induction with correction for multiple comparisons with a Holm–Sidak’s test: **** indicates an adjusted *p*-value < 0.0001, *** indicates *p* < 0.0001, ** indicates *p* < 0.001, while “ns” indicates that the adjusted *p*-value is not significant at α = 0.01.

**Figure 2 ijms-24-01513-f002:**
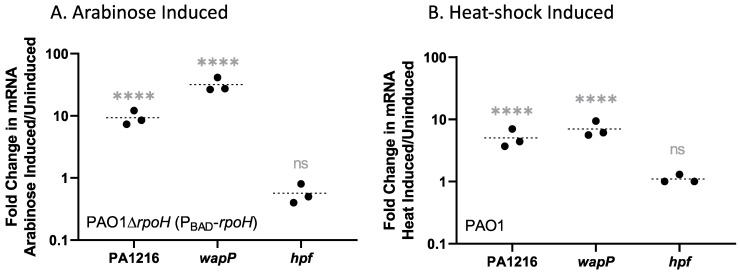
(**A**) Fold change of mRNA following 15 min of arabinose induction of *P. aeruginosa* ∆*rpoH* (P_BAD_-*rpoH*), showing significant differences of mRNA levels before and after arabinose treatment for PA1216 and *wapP*. (**B**) Fold change of PA1216 and *wapP* in wild-type strain *P. aeruginosa* PAO1, following 15 min of heat shock from 37 °C to 45 °C. No significant change was observed for the negative control mRNA *hpf* after either treatment. Three independent biological replicates were performed and the housekeeping gene *acpP* was used as an internal control. A two-way ANOVA was performed on delta Ct values pre and post induction with correction for multiple comparisons with a Holm–Sidak’s test: **** indicates an adjusted *p*-value < 0.0001 and “ns” indicates that the adjusted *p*-value is not significant at α = 0.01.

**Figure 3 ijms-24-01513-f003:**
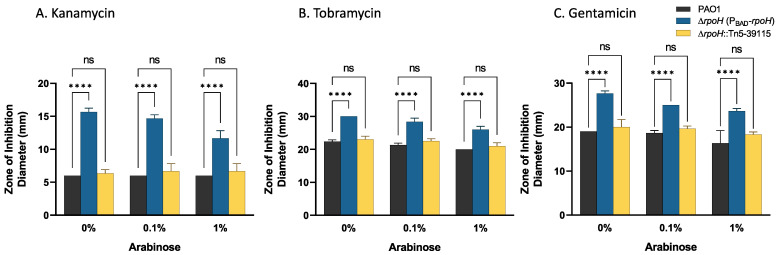
Antibiotic sensitivities of *P. aeruginosa* PAO1, *P. aeruginosa* ∆*rpoH* (P*_BAD_*-*rpoH*), and *P. aeruginosa rpoH*::Tn5-39115 using the disc diffusion assay. Strains were cultured in the absence of arabinose, then spread plated onto agar plates containing differing levels of arabinose and incubated with discs containing antibiotics. (**A**) kanamycin, (**B**) tobramycin, and (**C**) gentamicin. The results show the mean and standard deviation of the zone of inhibition (mm) for three biological replicates. The *P. aeruginosa* ∆*rpoH* (P*_BAD_*-*rpoH*) strain, but not the transposon mutant strain, showed a significant increase in sensitivity to the aminoglycoside antibiotics compared to the wild-type strain. Results of a two-way ANOVA corrected for multiple comparisons with a Holm–Sidak’s test are shown: **** indicates an adjusted *p*-value < 0.0001 while “ns” indicates that the adjusted *p*-value is not significant at α = 0.01.

**Figure 4 ijms-24-01513-f004:**
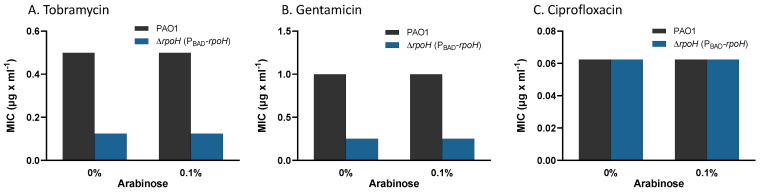
Antibiotic sensitivities of *P. aeruginosa* PAO1 and *P. aeruginosa* ∆*rpoH* (P*_BAD_*-*rpoH*) assayed as minimum inhibitory concentration (MIC) for (**A**) tobramycin, (**B**) gentamicin, and (**C**) ciprofloxacin. Three independent biological replicates assayed on a plate reader yielded identical results, as plotted above. The ∆*rpoH* (P*_BAD_*-*rpoH*) strain had increased susceptibility to the aminoglycosides tobramycin and gentamicin, but not to ciprofloxacin, when compared to the wild-type strain.

**Figure 5 ijms-24-01513-f005:**
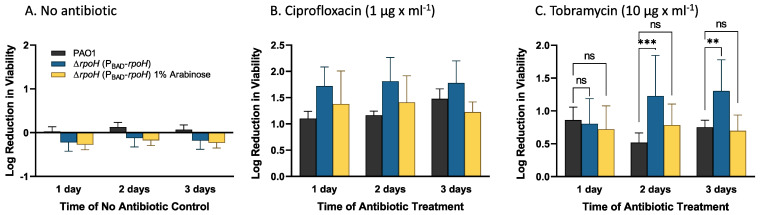
Antibiotic sensitivities of *P. aeruginosa* PAO1 and *P. aeruginosa* ∆*rpoH* (P*_BAD_*-*rpoH*) strains cultured in biofilms. Colony biofilms were cultured for 3 days and then exposed to ciprofloxacin or tobramycin for up to 3 additional days. Colony forming units were used to determine the log reduction of viable cells due to antibiotic treatment. (**A**) Control experiment to determine the effect of ∆*rpoH* deletion and arabinose addition on CFUs over the course of biofilm treatment time. (**B**) Reduction of viable cells in *P. aeruginosa* PAO1 and *P. aeruginosa* ∆*rpoH* (P*_BAD_*-*rpoH*) biofilms following exposure to 16X MIC for ciprofloxacin. No significant differences in log reduction were observed comparing *P. aeruginosa* PAO1 and *P. aeruginosa* ∆*rpoH* (P*_BAD_*-*rpoH*) over the course of three days of treatment by a two-way ANOVA with correction for multiple testing with a Holm–Sidak’s test at α = 0.01. (**C**) Reduction of viable cells in *P. aeruginosa* PAO1 and *P. aeruginosa* ∆*rpoH* (P*_BAD_*-*rpoH*) biofilms following tobramycin treatment at 20× MIC. No significant difference in log reduction was observed for the two strains following one day of antibiotic treatment, but the *P. aeruginosa* ∆*rpoH* (P*_BAD_*-*rpoH*) strain had a greater log reduction in CFU following after two and three days of treatment. Results of a two-way ANOVA corrected for multiple comparisons with a Holm–Sidak’s test are shown: *** indicates an adjusted *p*-value < 0.001, ** indicates an adjusted *p*-value < 0.01, and “ns” indicates that the adjusted *p*-value is not significant at α = 0.01. The mean and standard error for three independent biological replicates of the no antibiotic control experiments and four independent biological replicates of the antibiotic exposure experiments are plotted.

**Figure 6 ijms-24-01513-f006:**
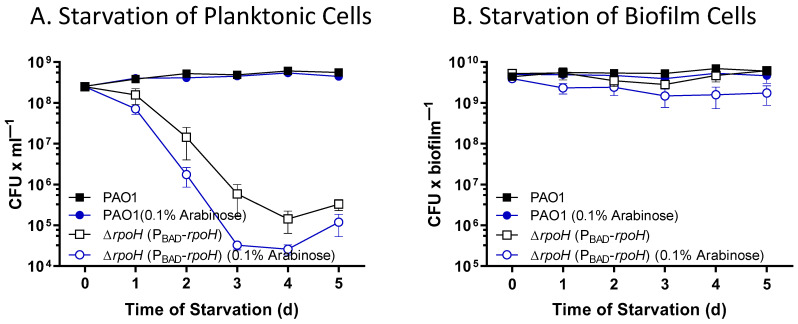
Effect of nutrient limitation on the viability of *P. aeruginosa* and *P. aeruginosa* ∆*rpoH* (P_BAD_-*rpoH*). Following growth to early stationary phase, cells were incubated in phosphate-buffered saline for up to 5 days. The viability of the cells was determined as colony forming units. (**A**) Starvation survival of *P. aeruginosa* cultured planktonically, showing no loss of viability of *P. aeruginosa* PAO1 (*p* > 0.99), but loss of viability for the *P. aeruginosa* ∆*rpoH* (P_BAD_-*rpoH*) strain (*p* < 0.0001). (**B**) Starvation survival of *P. aeruginosa* cultured in biofilms, showing little loss of viability for either strain. A minimum of three independent biological replicates were performed.

**Table 1 ijms-24-01513-t001:** Functional annotation clustering of *P. aeruginosa* ∆*rpoH* (P_BAD_-*rpoH*) genes significantly up or down regulated following 15 min of arabinose induction of *rpoH*.

Upregulated Gene Clusters *	*p*-Value
Oxidative phosphorylation	1.7 × 10^−9^
Stress response	3.9 × 10^−4^
Aerobic respiration	5.0 × 10^−3^
Chaperones	7.1 × 10^−3^
**Downregulated Gene Clusters ***	** *p* ** **-Value**
FecR	1.1 × 10^−5^
AMP-binding enzymes	1.5 × 10^−3^
TonB-dependent receptor	8.9 × 10^−3^

* Gene enrichment using DAVID at *alpha* < 0.01.

**Table 2 ijms-24-01513-t002:** Molecular chaperones, proteases, and accessory proteins significantly induced by a sudden increase in RpoH level.

ID ^1^	Gene	Function	Fold Change	*p*-Value	Essential ^2^
**PA0376**	** *rpoH* **	Heat shock sigma factor	16.6	1.4 × 10^−3^	+++
PA0778	*icp*	Inhibitor of cysteine peptidases	8.6	3.8 × 10^−6^	
**PA0779**	** *asrA* **	Stress response ATP-dependent protease	8.1	3.7 × 10^−4^	
**PA1596**	** *htpG* **	Heat shock protein—molecular chaperone	6.5	5.5 × 10^−7^	
PA1597		Dienelactone hydrolase/oligo peptidase	8.2	1.6 × 10^−4^	
**PA1802**	** *clpX* **	Protease	2.2	5.9 × 10^−3^	++
**PA1803**	** *lon* **	Protease	3.6	8.7 × 10^−3^	++
PA1805	*ppiD*	Peptidyl-prolyl cis-trans isomerase D	3.9	2.9 × 10^−4^	
PA2830	*htpX*	Zn-dependent protease/chaperone	3.1	6.2 × 10^−3^	
PA3126	*ibpA*	Chaperone	3.8	9.1 × 10^−4^	
PA3257	*algO*	Periplasmic protease	3.0	1.3 × 10^−3^	
PA3602		Glutamate synthase	2.4	3.7 × 10^−3^	
PA4472	*pmbA*	Zn-dependent protease	15.5	5.2 × 10^−5^	
**PA4474**		Zn-dependent protease	16.4	1.1 × 10^−3^	
PA4475		Amidohydrolase	5.0	4.0 × 10^−4^	
**PA4542**	** *clpB* **	Protease	10.8	4.8 × 10^−4^	
**PA4751**	** *ftsH* **	Membrane-bound, ATP-dependent protease	3.0	1.1 × 10^−4^	+++
PA4756	*carB*	Carbamoylphosphate synthase large subunit	2.2	1.7 × 10^−2^	++
PA4757		Amino acid efflux permease	2.9	1.1 × 10^−2^	
**PA4759**	** *dapB* **	Dihydropicolinate reductase	7.8	4.4 × 10^−3^	+++
**PA4760**	** *dnaJ* **	Chaperone	5.5	7.9 × 10^−4^	++
**PA4761**	** *dnaK* **	Chaperone	4.5	5.4 × 10^−3^	+++
**PA4762**	** *grpE* **	Chaperone	3.9	1.2 × 10^−3^	+
PA4858		Solute-binding sensor protein	2.1	4.0 × 10^−2^	
**PA4870**		DnaK suppressor; c4-type zinc finger protein	13.9	7.2 × 10^−4^	
**PA4942**	** *hflK* **	Inhibitor of protease activity	2.3	4.5 × 10^−2^	+
**PA4943**		GTPase	2.8	3.4 × 10^−3^	
**PA5053**	** *hslV* **	Protease	9.4	2.4 × 10^−4^	
**PA5054**	** *hslU* **	Protease	5.7	2.2 × 10^−4^	
PA5055		Similar to gamma-butyrobetaine dioxygenase	8.0	4.5 × 10^−4^	

^1^ Bold letters indicate overlap between genes found here and genes induced in *P. aeruginosa* PAO1 by heat shock (from ref. [25]). ^2^ + indicates that the gene was identified as essential under one condition in ref. [29], while “++” indicates it was essential under two conditions, and “+++” indicates that it was essential under all three conditions tested.

**Table 3 ijms-24-01513-t003:** Genes involved in amino acid metabolism that are significantly induced by a sudden increase in the level of RpoH.

ID	Gene	Function	Fold Change	*p*-Value	Essential ^1^
PA0392		Membrane protein	5.7	3.0 × 10^−4^	
PA0393	*proC*	Pyrroline-5-carboxylate reductase	7.5	5.2 × 10^−5^	+
PA0394		Tm barrel protein	2.3	1.5 × 10^−2^	
PA1742	*pauD2*	Glutamine amidotransferase	4.8	6.9 × 10^−4^	
PA1779		Assimilatory nitrate reductase	3.8	1.9 × 10^−2^	
PA1780	*nirD*	Assimilatory nitrite reductase small subunit	5.6	6.0 × 10^−3^	
PA1781	*nirB*	Assimilatory nitrite reductase large subunit	3.6	4.6 × 10^−2^	
PA1783	*nasA*	Nitrate transporter	4.0	3.0 × 10^−2^	
PA2776	*pauB3*	Arginine proline metabolism	4.8	1.0 × 10^−5^	

^1^ Essential genes as identified in ref. [29]. The “+” indicates that the gene was found to be essential for one of three conditions tested.

**Table 4 ijms-24-01513-t004:** Genes involved in nucleic acid metabolism that are significantly induced by a sudden increase in RpoH level.

ID ^1^	Gene	Function	Fold Change	*p*-Value	Essential ^2^
PA3010		Nucleotide binding domain	3.0	2.7 × 10^−3^	
**PA3011**	** *topA* **	Topoisomerase	5.4	6.7 × 10^−4^	+++
PA3272		ATP-dependent helicase	17.7	8.5 × 10^−5^	
PA4944	*hfq*	RNA binding	2.1	3.4 × 10^−3^	
PA4969	*cpdA*	3′,5′-cyclic-AMP phosphodiesterase	8.6	3.7 × 10^−4^	+++
PA4970		Conserved hypothetical protein	7.0	1.4 × 10^−3^	
PA4971	*aspP*	NTP pyrophosphohydrolase	11.9	6.1 × 10^−4^	

^1^ Bold letters show overlap between genes found here and genes induced in *P. aeruginosa* PAO1 by heat shock (from ref. [25]). ^2^ Essential genes identified in ref. [29]. The “+++” indicates that the genes were found to be essential for all the three conditions tested.

**Table 5 ijms-24-01513-t005:** Other genes significantly upregulated in PAO 1∆*rpoH* (P_BAD_-*rpoH*) by a 15 min induction of *rpoH*.

ID ^1^	Gene	Function	Fold Change	*p*-Value	Essential ^2^
PA0042		Hypothetical protein	2.1	3.3 × 10^−3^	
**PA0576**	** *rpoD* **	RNA polymerase sigma factor RpoD	2.7	2.0 × 10^−3^	+++
PA0744		Enoyl-CoA hydratase/isomerase	8.3	3.0 × 10^−4^	
PA0745	*dspI*	Enoyl-CoA hydratase/isomerase	10.7	6.2 × 10^−4^	
PA0746		Acyl-CoA dehydrogenase	10.5	1.8 × 10^−3^	
PA0747		NAD-dependent aldehyde dehydrogenase	14.2	6.3 × 10^−4^	
**PA0763**	** *mucA* **	Anti-sigma factor	2.1	2.5 × 10^−3^	+
PA0764	*mucB*	Negative regulator for alginate biosynthesis	2.9	1.9 × 10^−3^	+
**PA0905**	** *rsmA* **	CsrA carbon storage regulator	2.8	4.4 × 10^−2^	+
PA0916		Ribosomal protein S12 methylthiotransferase	2.0	3.2 × 10^−3^	
PA1211		α/β -hydrolase	2.1	1.3 × 10^−2^	
PA1212		Major facilitator superfamily; transporter	3.4	1.2 × 10^−2^	
PA1213		Gamma-butyrobetaine dioxygenase	4.5	4.5 × 10^−2^	
PA1214		Asparagine synthetase B	4.2	1.2 × 10^−2^	
PA1215		Acetyl-CoA synthetase; non-ribosomal peptide synthase	14.7	5.0 × 10^−3^	
PA1216		Isopropylmalate/homocitrate/citramalate synthase + methyltransferase	79.6	6.1 × 10^−5^	
PA1217		2-isopropylmalate or homocitrate synthase	47.9	3.1 × 10^−5^	
PA1218		2OG oxygenases that catalyze hydroxylation reactions	10.5	1.1 × 10^−4^	
PA1219		α/β-hydrolase	10.9	3.8 × 10^−3^	
PA1220		Condensation domain which makes peptide antibiotics	4.9	1.1 × 10^−2^	
PA1221		Non-ribosomal peptide synthase	5.3	2.0 × 10^−2^	
PA1318	*cyoB*	Cytochrome o ubiquinol oxidase, subunit I	2.2	9.2 × 10^−3^	
PA1319	*cyoC*	Cytochrome o ubiquinol oxidase, subunit III	2.1	6.3 × 10^−3^	
PA1555	*ccoP2*	Cytochrome c oxidase cbb3-type	4.0	7.0 × 10^−3^	
PA1557	*ccoN2*	Cytochrome c oxidase cbb3-type	3.4	4.5 × 10^−2^	
PA1550		Ig-like domain; also in FixH protein involved in cation pumping	2.1	5.1 × 10^−3^	
PA1826		LysR type regulator	2.3	1.4 × 10^−2^	
PA2101		Permease of the drug/metabolite transporters	4.3	5.9 × 10^−3^	
PA2102		Metal-dependent protease of the PAD1/JAB1 superfamily	3.8	2.2 × 10^−3^	
PA2103	*MoeB*	Molybdopterin biosynthesis MoeB protein	3.3	2.7 × 10^−3^	
PA2104		Cysteine synthase	3.2	1.9 × 10^−4^	
PA2106		Metal-dependent hydrolase, contains AlaS domain; alanyl-tRNA synthetase	2.2	2.2 × 10^−3^	
PA2275		Alcohol dehydrogenase	7.2	2.0 × 10^−4^	
PA2481		Cytochrome c oxidase, cbb3-type, subunit p	2.1	1.2 × 10^−2^	
PA2559		23 kda subunit of oxygen evolving system of photosystem II of plants	2.6	5.9 × 10^−3^	
PA2637	*nuoA*	NADH dehydrogenase I chain A	2.7	3.4 × 10^−3^	
PA2638	*nuoB*	NADH dehydrogenase I chain B	2.1	2.9 × 10^−2^	
PA2639	*nuoD*	NADH dehydrogenase I chain C,D	3.2	1.1 × 10^−2^	
PA2640	*nuoE*	NADH dehydrogenase I chain E	3.1	1.5 × 10^−2^	
PA2641	*nuoF*	NADH dehydrogenase I chain F	2.9	2.8 × 10^−3^	
PA2642	*nuoG*	NADH dehydrogenase I chain G	3.7	3.5 × 10^−3^	
PA2643	*nuoH*	NADH dehydrogenase I chain H	6.3	2.4 × 10^−3^	
PA2644	*nuoI*	NADH Dehydrogenase I chain I	2.9	9.6 × 10^−4^	
PA2645	*nuoJ*	NADH dehydrogenase I chain J	5.2	3.6 × 10^−4^	+
PA2646	*nuoK*	NADH dehydrogenase I chain K	3.9	1.8 × 10^−4^	
PA2647	*nuoL*	NADH dehydrogenase I chain L	4.4	2.5 × 10^−3^	
PA2648	*nuoM*	NADH dehydrogenase I chain M	3.6	1.5 × 10^−2^	
PA2649	*nuoN*	NADH dehydrogenase I chain N	5.4	1.3 × 10^−3^	
PA2811		ABC transporter, permease component	6.2	1.6 × 10^−3^	
PA2812		ABC transporter, ATP-binding component	5.7	5.1 × 10^−5^	
PA3136		Involved in substrate efflux	2.0	1.1 × 10^−2^	
PA3472		Invasion protein; cell wall-associated hydrolase degrades peptidoglycans	2.4	4.2 × 10^−2^	
PA3531	*bfrB*	Bacterioferritin	6.6	7.1 × 10^−4^	
PA3950		ATP-dependent helicase	2.6	5.5 × 10^−4^	
PA3951		PIN-domain nuclease	13.7	2.8 × 10^−3^	
PA3952		Hypothetical protein	16.2	7.9 × 10^−4^	
PA3979		Similarity to stringent starvation protein	2.0	5.6 × 10^−3^	
**PA4061**	** *ybbN* **	Thioredoxin	7.8	1.2 × 10^−4^	
PA4333		Fumarase	2.0	4.1 × 10^−3^	+++
PA4387		FxsA protein affecting phage T7 exclusion by the F plasmid	10.4	6.7 × 10^−6^	
PA4428	*sspA*	Stringent response protein	3.0	3.2 × 10^−2^	
PA4429		Cytochrome c1 precursor	2.3	5.4 × 10^−3^	+++
PA4430		Cytochrome b	2.7	2.7 × 10^−3^	+
PA4503	*dppB*	ABC transporter, permease component; nickel, sulfate or small peptides	2.0	3.5 × 10^−2^	
PA4504	*dppC*	ABC transporter, permease component	2.2	3.2 × 10^−2^	
PA4749	*glmM*	Phosphoglucosamine mutase	2.3	6.3 × 10^−4^	+++
PA4750	*folP*	Dihydropteroate synthase	3.8	3.2 × 10^−3^	+++
PA4812	*fdnG*	Formate dehydrogenase-O, major subunit	2.1	3.4 × 10^−2^	
PA4839	*speA*	Arginine decarboxylase	3.0	3.8 × 10^−4^	
PA4840		Putative translation initiation factor SUI1	3.9	1.4 × 10^−5^	
PA4841		Nudix hydrolase	15.6	1.2 × 10^−4^	
PA4842		Hypothetical protein	7.2	2.4 × 10^−3^	
PA4862		ABC transporter, ATP-binding subunit	3.1	3.0 × 10^−2^	
PA4865	*ureA*	Urease gamma subunit	2.7	1.5 × 10^−2^	
PA4866		Urease complex protein	2.1	3.1 × 10^−2^	
PA4888	*desB*	Acyl-CoA delta-9-desaturasE	2.9	1.6 × 10^−2^	
PA4889		Dehydrogenase/reductase	2.2	2.9 × 10^−2^	
PA4909		ABC transporter ATP-binding subunit	2.4	2.1 × 10^−2^	
PA4910		ABC transporter ATP-binding subunit	3.2	1.9 × 10^−2^	
PA4912		Branched-chain amino acid ABC transporter, permease subunit	2.4	2.5 × 10^−2^	
PA4913		Periplasmic; branched-chain amino acid ABC transporter	2.8	1.5 × 10^−2^	
PA5001	*ssg*	Cell surface-sugar biosynthetic glycosyltransferase	3.0	3.9 × 10^−3^	+
PA5002	*dnpA*	De-N-acetylase involved in persistence	2.5	1.8 × 10^−2^	
PA5003		Mig-14-like protein	3.3	3.3 × 10^−3^	+
PA5004	*wapH*	Glycosyltransferase, group 1	4.1	9.3 × 10^−3^	
**PA5005**		Carbamoyl transferase	3.9	1.7 × 10^−3^	
PA5006		Serine/threonine-protein kinase	7.6	1.2 × 10^−3^	+++
PA5007	*wapG*	Serine/threonine-protein kinase	10.8	1.3 × 10^−4^	
PA5008	*wapP*	Serine/threonine-protein kinase	8.9	1.3 × 10^−5^	+++
PA5009	*waaP*	Lipopolysaccharide kinase	11.7	9.5 × 10^−5^	+++
**PA5010**	** *waaG* **	UDP-glucose:(heptosyl) LPS alpha 1,3-glucosyltransferase	8.7	1.9 × 10^−5^	+++
PA5124PA5124	*ntrB*	Two-component sensor	2.2	1.8 × 10^−2^	
PA5159		Multidrug resistance efflux pump	2.7	2.3 × 10^−2^	
PA5160		Drug efflux transporter	2.0	4.1 × 10^−2^	
PA5203	*gshA*	Glutamate--cysteine ligase	3.7	5.2 × 10^−3^	+
PA5380	*gbdR*	Transcriptional regulator containing an amidase domain and an AraC-type DNA-binding HTH domain	2.0	4.0 × 10^−2^	
PA5479	*gltP*	Proton-glutamate symporter	2.0	7.6 × 10^−4^	

^1^ Bold letters indicate overlap between genes found here and genes induced in *P. aeruginosa* PAO1 by heat shock (from ref [25]). ^2^ Essential genes identified in ref [29]. The number of “+” is the number of conditions where the gene was found to be essential of the three conditions tested.

**Table 6 ijms-24-01513-t006:** Overlap of genes significantly affected by RpoH induction with genes of other functional classes.

	*p* Value for Overlap with Genes Significantly:	
Activity ^1^	Upregulated by *rpoH* Induction	Downregulated by *rpoH* Induction
**Heat shock** (ref. [25])	** *p* ** ** < 1 × 10^−15^**	*p* > 0.999
**Heat shock** (pseudomonas.com)	** *p* ** ** = 7.7 × 10^−6^**	*p* > 0.999
**Planktonic ciproflaxacin tolerance**	***p* = 0.001**	*p* = 0.510
**MvfR+**	** *p* ** ** = 0.009**	*p* = 0.215
**Mouse wound**	** *p* ** ** = 0.031**	*p* = 0.033
**Planktonic tobramycin sensitivity**	** *p* ** ** = 0.034**	*p* = 0.445
**Iron limitation**	*p* = 0.789	** *p* ** **< 1 × 10^−15^**
**Stationary phase**	*p* = 0.893	** *p* ** **< 1 × 10^−15^**
**RpoS+**	*p* = 0.127	** *p* ** ** = 2.3 × 10^−15^**
**Virulence**	*p* = 0.993	** *p* ** ** = 7.3 × 10^−8^**
**Peroxide stress**	*p* = 0.352	** *p* ** ** = 0.003**
**Biofilm**	*p* = 0.062	** *p* ** ** = 0.007**
HLS QS	*p* > 0.999	*p* = 0.442
Mg limitation	*p* = 0.443	*p* > 0.999
Phenazine biosynthesis	*p* > 0.999	*p* > 0.999
Efflux pumps	*p* > 0.999	*p* > 0.999
Nitrosative stress	*p* > 0.999	*p* > 0.999
Planktonic ciproflaxacin sensitivity	*p* = 0.228	*p* > 0.999
Up in biofilm treated with ciprofloxacin	*p* > 0.999	*p* = 0.807
Up in biofilm treated with tobramycin	*p* = 0.551	*p* = 0.675
SOS response	*p* > 0.999	*p* > 0.999
Zn limitation	*p* > 0.999	*p* > 0.999
Anr+	*p* = 0.732	*p* = 0.536
Crc+	*p* > 0.999	*p* > 0.999

^1^ Lists of genes associated with each activity are from refs. [16,25] and from pseudomonas.com, accessed on 2 February 2022.

## Data Availability

Microarray data have been deposited in the NCBI Gene Expression Omnibus and are accessible through GEO series accession number GSE217157.

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
