# Peer review of "The Pseudomonas aeruginosa RpoH (σ32) Regulon and Its Role in Essential Cellular Functions, Starvation Survival, and Antibiotic Tolerance"

_ijms, 2023, doi:10.3390/ijms24021513_

Round 1

Reviewer 1 Report

The manuscript by Williamson and co-authors presents data on the effect of rpoH gene deletion on Pseudomonas and the observed effects on cell viability.

Minor changes should be made in order to increase manuscripts quality.

1. All figures must be on high quality - please double check Figs 5 and 6.

2. Please standardize the restriction enzymes names. The first 3 letters should be in italics, e.g. EcoRI.

3. Did you considered using STRING to analyze the genes co-expressed with rpoH?

4. L 153. Please clarify why/how 7.75 h was set as the induction time point.

5. L 228. Table 2 - genes should be in italics. Please correct.

6. Same for tables 3, 4, 5.

7. L 650-654. Clarify if your plasmid contain the AraC encoding sequence for regulation of the expression.

8. From your results, could RpoH be a terapeutic target for the treatment of Pseudomonas infections?

Reviewer 2 Report

In this manuscript Kerry S. Williamson and colleagues aimed to characterize genes under the direct regulation of a sigma 32 factor RpoH in the opportunistic human pathogen Pseudomonas aeruginosa. This protein is known to be involved in response to accumulation of misfolded proteins through regulation of gene expression. Since response to stresses inducing protein (e.g. heat-shock) is complex, here the authors used a strain in which rpoH was placed under the control of arabinose induce promoter PBAD, to dissect the genes under the direct control of this factor. Moreover in this manuscript, a function of RpoH in response to aminoglycosides and survival in biofilms is also proposed. Overall the study is clearly described, presented results are supported by statistical analyses, and overall provide a significant advance in the understanding of RpoH functioning in P. aeruginosa. Some points must be however clarified before publication.

Major comments

1. Throughout the manuscript the strain ΔrpoH(PBAD-rpoH) is compared to PAO1. The ΔrpoH(PBAD-rpoH) carries a plasmid, whereas the ‘PAO1’, as I understand, is not a strain with the corresponding empty vector (pCF430). The presence of plasmid (empty vector) alone might influence the growth and physiological parameters of the cells (even when grown without selection). The effect of the plasmid presence should be ruled out particularly in the experiments presented on Fig 4,5,6.

2. Line 195 , line 681 - In the transcriptome analyses the authors do not consider the possibility that the addition of arabinose alone might change expression of some genes. The process of adding the inducer (taking out the cultures, opening, adding arabinose) might also have an effect on gene expression. Have these factors been ruled out, e.g. by addition of water to control cultures? Theses limitations (effect of arabinose, opening of cultures etc.) should at least be discussed the text as they might cause some of the gene expression changes.

3. Based on the list of genes directly regulated by RpoH, can the authors analyze their promoters and check whether the -35 and -10 promoter consensus sequences are the same for Pseudomonas aeruginosa RpoH and other species?

Minor remarks (data presentation, details, comments)

1. line 20 -“whole genome transcriptome” - transcriptome analyses are usually global so no need for whole genome here. Please check this sentence as to me it reads like the transcriptome analysis also showed that many of the genes are essential (second part of sentence) which is not true.

2. Line 45 – accumulation is what activates the response and is not a part of the response itself

3. Line 97 - Data from Poulsen et al., 2019 https://doi.org/10.1073/pnas.1900570116 could also be used to validate the statement that rpoH is essential (although in that study PAO1 wasn't included, it should be possible to look at the orthologs in other strains)

4. Line 148 – leaky PBAD was described by Meisner J and Goldberg JB. 2016. Appl Environ Microbiol 82:6715– 6727. doi:10.1128/AEM.02041-16. Might be good to include this or other reference as to some researchers find it surprising that common promoters are leaking in P. aeruginosa cells and there is no inducer needed to see an effect.

5. line 153 – time is given, but to what is the usual OD 600 nm / stage of growth? The authors might consider to add a reference to a published growth curve or include one in the supplement.

6. Line 184 – Figure 1 has no caption

7. e.g. line 188 shouldn't a paired T-test be used here? I am not sure if ANOVA is applicable if there are only two groups.

8. for clarity please include P. aeruginosa PA…. gene IDs, on the first time a gene is mentioned in the text. For P. aeruginosa the common gene names change between the databases and various annotations.

9. Line 249, Line 275-277 and 833-840. How is the outcome of these analyses used in the manuscript? i.e. please elaborate on how the outcome of these analyses was used for prediction of protein function

10 Please use RCF rather than RPM in the Materials & methods

11. line 23 – damage and repair → without and?; line 50 – s32; line 41 pH → ‘extreme’ pH or ‘pH changes’; line 106 – problem with reference; line 216 – high-throughput transposon library → delete high-throughput?

The ideal report should be organized as follows:

  • An introductory paragraph placing the manuscript in a broader context, describing the contribution of the manuscript to the field, summarizing the manuscript's main findings and claims, and giving the Referee's overall impression of the manuscript.

  • Specific comments which should be further divided into Major and Minor Comments.

Suggestions to improve the manuscript. These may include:

  • new experiments or improvements to the described experiments

  • addition/deletion of references

  • changes to the text to improve presentation, quality of English, length etc.

Round 2

Reviewer 2 Report

Thank you for addressing my comments -  the text changes, data added, and the responses are satisfactory. Congratulations on the good work!